# AAV-SPL 2.0, a Modified Adeno-Associated Virus Gene Therapy Agent for the Treatment of Sphingosine Phosphate Lyase Insufficiency Syndrome

**DOI:** 10.3390/ijms242115560

**Published:** 2023-10-25

**Authors:** Ranjha Khan, Babak Oskouian, Joanna Y. Lee, Jeffrey B. Hodgin, Yingbao Yang, Gizachew Tassew, Julie D. Saba

**Affiliations:** 1Department of Pediatrics, Division of Hematology/Oncology, University of California, San Francisco, CA 94143, USA; ranjha.khan@ucsf.edu (R.K.); babak.oskouian@ucsf.edu (B.O.); joanna.lee@ucsf.edu (J.Y.L.); gizachew.tassew@ucsf.edu (G.T.); 2Department of Pathology, University of Michigan School of Medicine, Ann Arbor, MI 48109, USA; jhodgin@med.umich.edu (J.B.H.); yyingbao@med.umich.edu (Y.Y.)

**Keywords:** sphingosine phosphate lyase insufficiency syndrome, *Sgpl1*, sphingolipid, gene therapy, AAV-SPL 2.0, adeno-associated virus, inborn-error of metabolism

## Abstract

Sphingosine-1-phosphate lyase insufficiency syndrome (SPLIS) is an inborn error of metabolism caused by inactivating mutations in *SGPL1*, the gene encoding sphingosine-1-phosphate lyase (SPL), an essential enzyme needed to degrade sphingolipids. SPLIS features include glomerulosclerosis, adrenal insufficiency, neurological defects, ichthyosis, and immune deficiency. Currently, there is no cure for SPLIS, and severely affected patients often die in the first years of life. We reported that adeno-associated virus (AAV) 9-mediated *SGPL1* gene therapy (AAV-SPL) given to newborn *Sgpl1* knockout mice that model SPLIS and die in the first few weeks of life prolonged their survival to 4.5 months and prevented or delayed the onset of SPLIS phenotypes. In this study, we tested the efficacy of a modified AAV-SPL, which we call AAV-SPL 2.0, in which the original cytomegalovirus (CMV) promoter driving the transgene is replaced with the synthetic “CAG” promoter used in several clinically approved gene therapy agents. AAV-SPL 2.0 infection of human embryonic kidney (HEK) cells led to 30% higher SPL expression and enzyme activity compared to AAV-SPL. Newborn *Sgpl1* knockout mice receiving AAV-SPL 2.0 survived ≥ 5 months and showed normal neurodevelopment, 85% of normal weight gain over the first four months, and delayed onset of proteinuria. Over time, treated mice developed nephrosis and glomerulosclerosis, which likely resulted in their demise. Our overall findings show that AAV-SPL 2.0 performs equal to or better than AAV-SPL. However, improved kidney targeting may be necessary to achieve maximally optimized gene therapy as a potentially lifesaving SPLIS treatment.

## 1. Introduction

Sphingosine-1-phosphate lyase insufficiency syndrome (SPLIS), also known as nephrotic syndrome type 14 (NPHS14), is an autosomal recessive inborn error of metabolism first reported in 2017 [1,2]. The condition is caused by bi-allelic inactivating mutations in *SGPL1*. *SGPL1* encodes sphingosine-1-phosphate lyase (SPL), an essential enzyme needed to degrade sphingolipids [3]. The enzyme requires a pyridoxal 5′-phosphate cofactor and catalyzes the irreversible cleavage of the bioactive lipid sphingosine-1-phosphate (S1P), resulting in the formation of hexadecenal and ethanolamine phosphate [4]. Many variants of *SGPL1* have been reported [5,6]. SPLIS-associated *SGPL1* mutations lead to missing or nonfunctional SPL protein and insufficient SPL enzyme activity, which causes the failure of the sphingolipid metabolism. SPL functions at the only exit point of the sphingolipid metabolism. Thus, its inactivation leads to the accumulation of the fibrogenic and pro-inflammatory signaling molecule S1P and other phosphorylated long-chain bases, as well as upstream sphingolipid intermediates such as ceramides and sphingosine, which are growth inhibitory and cytotoxic molecules [7]. The products of the SPL reaction are important contributors to cellular functions such as autophagy, and their deficiency in SPLIS may also lead to pathology [8]. Affected children have a wide range of presentations with one or more of the main disease features, which include a rapidly progressive form of glomerulosclerosis, primary adrenal insufficiency, central and/or peripheral neurological defects, ichthyosis, and lymphopenia with or without immune deficiency. Currently, there is no curative or effective therapy for SPLIS. Severely affected patients who harbor totally non-functional mutants of SPL often die in the first year of life due to kidney failure and general wasting or infectious causes. Treatment is supportive and may include the correction of glucose and electrolyte imbalance, hormone supplementation, anti-epileptic drugs, physical therapy, kidney dialysis, and kidney transplantation.

Gene therapy represents a potentially lifesaving treatment strategy for patients with inborn errors of metabolism and other monogenic diseases [9,10]. It acts by correcting the root cause of the disease, providing a healthy, functional copy of the affected gene to key target tissues where the gene is normally expressed and needed to sustain tissue or organ function. Ideally, the treatment is instituted during a window of opportunity between the time of diagnosis and the time of irreversible tissue damage and/or organ failure. Although many viral vectors have been used in gene therapy preclinical studies, AAV is considered a nonpathogenic virus in humans, is well-characterized, and has the advantage of being less prone to DNA integration [11,12]. It has been tested in over a hundred clinical trials to date and is used as the vector in several clinically approved gene therapies [9]. Despite the promise of AAV gene therapy, severe complications and deaths have been reported in association with its use in some recent clinical trials [13]. Therefore, it is critical to optimize gene therapies in development for use in humans to achieve the desired effects with the lowest possible dose.

We recently reported the efficacy of an AAV-based gene therapy strategy for SPLIS [14]. We found that adeno-associated virus 9-mediated *SGPL1* gene therapy (AAV-SPL) given to newborn *Sgpl1* knockout mice that model SPLIS and die in the first few weeks of life prolonged their survival to 4.5 months and prevented or delayed the onset of SPLIS phenotypes. However, the dose required for efficacy has been associated with severe toxicity and deaths in some human clinical trials [13,15,16]. Furthermore, the cytomegalovirus (CMV) promoter used to drive the *SGPL1* transgene has limitations, including its susceptibility to epigenetic silencing, immunogenicity, nonspecific cell toxicity, and a risk of reactivating latent CMV infection [17,18]. Therefore, AAV-SPL optimization is necessary before further development can proceed. 

In this study, we tested the efficacy of a modified AAV-SPL, which we call AAV-SPL 2.0, in which the original CMV promoter driving the transgene is replaced with the synthetic “CAG” promoter used in several clinically approved gene therapy agents. Our results show that modification of the vector by introducing a more suitable promoter not only did not compromise efficacy but also showed superiority to the original virus.

## 2. Results

### 2.1. AAV-SPL 2.0 Results in Higher SPL Expression and Activity Than AAV-SPL after Infection of Human Kidney Cells

To develop a safer, more efficient gene therapy agent for the treatment of SPLIS, we replaced the CMV promoter in our AAV-SPL vector. The new construct uses the synthetic CAG promoter to drive the expression of the human *SGPL1* cDNA. Our construction strategy is outlined in the Section 4. CAG contains sequences of the cytomegalovirus CMV early enhancer element, the promoter, first exon and first intron of the chicken beta-actin gene, and the splice acceptor of the rabbit beta-globin gene [19,20]. The CAG promoter is an element of Zolgensma, an AAV gene therapy agent approved for the treatment of spinal muscular atrophy by the Food and Drug Administration. The modified vector was named AAV-SPL 2.0 (Figure 1A). Both AAV-SPL and AAV-SPL 2.0 were packaged in the AAV9 capsid, which was chosen for its broad tropism, which includes the central and peripheral nervous system and adrenal gland, with some activity in the kidney, and its demonstrated safety and efficacy in clinical trials. When compared head-to-head in vitro by transfection of HEK293T cells with 100–500 ng AAV-SPL vs. AAV-SPL 2.0 plasmid DNA, AAV-SPL 2.0 resulted in approximately 30% higher SPL protein expression (Figure 1B) and enzyme activity (Figure 1C) compared to AAV-SPL. Based on these results, we proceeded to test the in vivo efficacy of AAV-SPL 2.0.

### 2.2. AAV-SPL 2.0 Promotes the Growth and Survival of Sgpl1 KO Mice

*Sgpl1* KO mice exhibit runting by the first week of life and die around the time of weaning in the third week of life [21]. Their short lifespans make these mice an ideal model for establishing AAV-SPL 2.0’s impact on SPLIS survival. Our previous studies revealed that AAV-SPL gene therapy could prolong the survival of *Sgpl1* KO mice if administered in the first few days of life but not if given at two weeks of age [15]. Therefore, we administered AAV-SPL 2.0 in the same manner by genotyping litters of heterozygous matings at birth (Figure 2A), followed by injecting the virus intravenously at a dose of 5 × 10^11^ vg within the first 5 days of life. As shown in Figure 2B, green food coloring added to the virus solution allows for the discrimination of unsuccessfully injected pups (a, top) from a successful injection (b, bottom). Compared to the runted, untreated *Sgpl1* KO mice, KO pups treated with AAV-SPL 2.0 looked healthy and gained weight over time (Figure 2C,D). Some mice were euthanized at 26 days of life (DOL) for bioavailability studies. The kidneys of untreated KO mice were smaller, paler, and weighed less than those of wild-type (WT) and treated KO mice (Figure 2E,F). 

Compared to untreated *Sgpl1* KO pups, none of which survived the perinatal period, *Sgpl1* KO pups treated with AAV-SPL 2.0 lived between five and eight months, with a mean survival of 148 days ± 48, representing improvement over no treatment (i.e., a life span of 13 ± 3 days) and our historical results with AAV-SPL (137 ± 114 days) (Figure 2G). The treated KO mice followed a similar growth pattern as WT mice, gaining weight steadily during the first four months of life (Figure 2H). Although their rate of growth was similar to WT mice, the absolute body weights of treated KO mice were significantly lower than those of WT mice, and after the age of four months, the body weights of treated KO mice showed a steeper decline than those of WT mice. In contrast to the positive effects of AAV-SPL 2.0 on *Sgpl1* KO mice weight gain and survival, KO mice receiving an AAV-GFP control virus showed no improvement over untreated KO mice (Figure 2G,H).

### 2.3. AAV-SPL 2.0 Prevents Neurodevelopmental Delay in Sgpl1 KO Mice

We previously reported that *Sgpl1* KO mice exhibit delayed neurodevelopment and impaired motor function compared to WT littermates during the first few weeks of life [14]. Using the same battery of neurodevelopmental tests designed for pre-weaned pups, we confirmed that untreated *Sgpl1* KO mice exhibit a delay in achieving five key neurodevelopmental milestones (Figure 3A–F). However, the treatment of *Sgpl1* KO mice with AAV-SPL 2.0 restored normal neurodevelopment and motor function (as measured by grip strength), which was not significantly different between treated *Sgpl1* KO mice and WT controls. In contrast, the development and strength of *Sgpl1* KO mice treated with an AAV-GFP control virus exhibited a neurodevelopmental delay similar to untreated KO mice (Figure 3A–F). 

### 2.4. AAV-SPL 2.0 Delays Onset of Nephrotic Syndrome and Glomerulosclerosis in Sgpl1 KO Mice

*Sgpl1* KO mice exhibit massive proteinuria, as measured by high urine albumin/creatinine ratios (ACR) prior to their demise at the time of weaning [14]. Histologically, the kidney cortices of KO mice show enlarged glomeruli with mesangial expansion, and ultrastructural analysis demonstrates podocyte foot process effacement, the classic sign of nephrotic syndrome. We reported that these abnormalities were normalized or attenuated by AAV-SPL treatment [15]. In the present study, untreated *Sgpl1* KO mice demonstrated a urine ACR level of 66.06 ± 3.80 prior to their demise at 3 weeks of age. In contrast, the urine ACR levels of WT mice were 11.58 ± 2.57 over the five months of observation (Figure 4A). The urine ACR levels of AAV-SPL 2.0-treated KO mice at the time point of 2 months and 3 months of age were 19.06 ± 0.30 and 17.6 ± 5.11, respectively, but began to increase at 4 months of age (46.60 ± 7.21), although they remained lower than those of untreated KO mice. However, at five months of age, the urine ACR levels of the AAV-SPL 2.0-treated KO mice were 163.17 ± 50.42, surpassing the levels of untreated KO mice, indicating a massive failure of glomerular function (Figure 4A). Consistent with these findings, Periodic acid Schiff-stained fixed kidney sections of untreated *Sgpl1* KO mice exhibited signs of glomerulosclerosis at 26 days of life, including mesangial expansion, with a mean of 18% sclerotic glomeruli compared to no detectible sclerotic glomeruli in either WT or AAV-SPL 2.0-treated mouse kidneys (Figure 4B, left panel). Furthermore, immunohistochemical staining of fixed kidney sections to detect activated (phosphorylated) STAT3, a marker of glomerular injury, showed a high number of signal in the kidneys of KO mice, whereas no STAT3 activation was detected in WT kidneys and very few signals were detected in AAV-SPL 2.0-treated KO kidneys at 26 days of life (Figure 4B, right panel). These results indicate that AAV-SPL 2.0 prevented the early onset of the glomerular disease characteristic of the *Sgpl1* KO mouse. No untreated KO mice were alive at 5 months of age. However, when the kidneys of WT mice and SPL 2.0-treated KO mice were compared at 5 months of age, the treated KO mouse kidneys showed diffuse mesangial expansion, a mean of 11% sclerotic glomeruli, severe interstitial fibrosis, scattered protein casts, and signs of acute tubular injury (Figure 4C, left panel). Similarly, IHC staining for activated STAT3 revealed diffuse nuclear positivity among tubules and glomeruli, and probably immune cells (Figure 4C, right panel). These cumulative findings suggest that AAV-SPL 2.0 treatment can delay but not completely prevent the eventual onset of nephrosis and kidney pathology in *Sgpl1* KO mice.

Podocytes are epithelial cells of the kidney glomerulus responsible for filtering the blood. Podocytes line the basement membrane of the glomerular capillaries and filter the flow through the fenestrated endothelial cells via the actions of a multiprotein complex called the slit diaphragm positioned between the podocyte foot processes [22]. The slit diaphragm discriminates between the plasma proteins of small and large molecular weight, retaining the large proteins and allowing the small ones to pass into the pre-urine. The damage of podocyte foot processes and loss of slit diaphragm proteins such as nephrin, podocin, and ancillary proteins such as synaptopodin are thought to give rise to the nephrotic syndrome [23]. To compare the status of podocytes in treated and untreated *Sgpl1* KO and WT mouse kidneys, we performed the Western blotting of whole kidney extracts to compare the abundance of nephrin, synaptopodin, and beta-actin as a loading control. As shown in Figure 4D, untreated KO mouse kidneys expressed almost no detectible nephrin and a marked reduction in synaptopodin levels in comparison to WT kidneys, indicating a profound loss of podocytes in the KO. In comparison, nephrin levels in KO mice treated with AAV-SPL 2.0 were expressed at the level of WT, and synaptopodin levels were markedly higher in treated compared to untreated KO kidneys.

### 2.5. Other Features of SPL Insufficiency Are Improved by AAV-SPL 2.0 in Sgpl1 KO Mice

In addition to improving growth and survival and delaying the progression of kidney disease, treatment with AAV-SPL 2.0 attenuated other features of SPL insufficiency, including anemia (as shown by a rise in hemoglobin, hematocrit, and red blood cell mass) and a doubling of the absolute lymphocyte count and percent of leukocytes that are lymphocytes (Figure 5A–D). Interestingly, in our previous study, AAV-SPL did not impact the lymphopenia of *Sgpl1* KO mice [15]; thus, our findings reveal an important benefit of AAV-SPL 2.0.

### 2.6. Pro-Inflammatory and Pro-Fibrogenic Signals Are Attenuated in Sgpl1 KO Mouse Kidneys by Treatment with AAV-SPL 2.0

The SPL substrate S1P and the pro-inflammatory transcription factor STAT3 stimulate mutually co-activating pathways that can drive inflammatory processes, including diabetic kidney disease [24,25,26]. Further, STAT3 is activated in *Sgpl1* KO mouse kidneys (Figure 4C) [14]. The treatment of *Sgpl1* KO mice with AAV-SPL 2.0 resulted in a dramatic reduction in kidney STAT3 activation, as demonstrated by the reduced detection of phospho-STAT3 by Western blotting of whole kidney tissue extracts of AAV-SPL 2.0-treated KO mice at 23 DOL (Figure 6A, and quantified in Figure 6B). Furthermore, downstream STAT3 transcriptional target genes including *Lcn2*, *Timp1*, *Socs1,* and *Socs3* were downregulated by the AAV-SPL 2.0 treatment, as determined by quantitative RT-PCR (Figure 6C–F). Additional cytokines, including *Mcp1* and *Tnf-α*, were also downregulated in the kidneys of *Sgpl1* KO mice that had received the treatment (Figure 6G,H). Lastly, the fibrotic protein smooth muscle actin (SMA) encoded by *Acta2* was downregulated by the AAV-SPL 2.0 treatment of KO mice (Figure 6I). Altogether, these findings demonstrate a significant blunting of pro-inflammatory and fibrogenic signals in the kidneys of KO mice treated with AAV-SPL 2.0.

### 2.7. AAV-SPL 2.0 Treatment Restores SPL Expression and Activity in Tissues of Sgpl1 KO Mice

Using quantitative RT-PCR, we compared *SGPL1* expression levels in various tissues. Consistent with the known tropism of AAV9, AAV-SPL 2.0-mediated hSPL (*SGPL1*) expression was found in the heart, muscle, liver, and lung, and less in other tissues (Figure 7A). As shown in Figure 7B, murine *Sgpl1* expression levels are highest in the intestine and thymus, tissues where its function in the metabolism of dietary sphingolipids and regulation of lymphocyte egress are well known. As shown in Figure 7C, SPL activity levels in AAV-SPL 2.0-treated KO kidneys and liver were comparable to WT levels, ranging from 24 to 32 pmol/mg/min in the kidney. Consistent with these findings, the tissue levels of S1P in the kidneys of WT and AAV-SPL-2.0 treated KO mice were much lower than in untreated KO kidneys, revealing an accumulation of the SPL substrate in the absence of SPL activity in the KO, which was corrected by gene therapy (Figure 7D). 

### 2.8. Efficacy of AAV-SPL 2.0 Is Lost upon Reduction in the Dose Lower than 5 × 10^11^ vg

By replacing the CMV promoter with the CAG promoter, without compromising efficacy, AAV-SPL 2.0 achieved a significant goal of our study, which was to develop a suitable biologic agent for translation into humans. To determine the dose needed for efficacy, a dose range-finding survival study was conducted by treating *Sgpl1* KO mice with AAV-SPL 2.0 at doses ranging from 1 × 10^10^ vg to 1 × 10^12^ vg in half-log increments. As shown in Figure 8, doses at or above 5 × 10^11^ vg, which translates to 3 × 10^14^ vg/kg when given in a 1.5 g pup, were efficacious in prolonging the survival of *Sgpl1* KO mice. Thus, the replacement of the CMV promoter with a CAG promoter maintained the potency of our original virus. 

## 3. Discussion

SPLIS is a rare and lethal childhood syndrome that was first recognized 8 years ago [1]. SPLIS is manifested by various features, such as steroid-resistant nephrotic syndrome (SRNS), leading to the rapid development of end-stage renal disease, adrenal insufficiency, and neurological defects. Various targeted therapeutic approaches, including enzyme replacement, gene therapy, and gene editing, have the potential to fully restore SPL activity in patients with SPLIS. However, the efficacy of enzyme replacement therapy in SPLIS remains uncertain due to the fact that SPL is normally located in the endoplasmic reticulum membrane, whereas the typical target site of enzyme replacement therapy is the lysosome. Furthermore, although CRISPR/Cas9 genome editing technology is developing rapidly, problems with off-target effects and efficient systemic delivery are major obstacles that remain to be solved. A gene therapy that delivers a healthy, active *SGPL1* gene to critical tissues may represent a near-term solution that can address the root cause of SPLIS by providing SPL function where it is needed to generate S1P gradients, prevent the accumulation of toxic sphingolipids, and generate bioactive molecular products, thereby preventing the key features of SPLIS. Our first-generation AAV-SPL showed efficacy in the *Sgpl1* KO mouse model of SPLIS. However, it had several drawbacks, including the presence of a viral promoter element (CMV), lack of any observed impact on lymphopenia, and the high dose required to achieve prolongation of animal survival. 

In an attempt to address the issues that we noted in our first-generation AAV-SPL, we created a second-generation AAV-SPL agent, AAV-SPL 2.0, in which the human *SGPL1* transgene is driven by CAG, a strong constitutive promoter that is present in a number of FDA-approved gene therapies. In vitro, we observed a 30% increase in SPL expression and activity after infection of HEK293 cells with AAV-SPL 2.0 compared to AAV-SPL. In vivo, AAV-SPL 2.0 treatment resulted in a consistent extension of the survival of *Sgpl1* KO mice from several weeks to 5 months, achieving a mean survival of 5.6 months, an improvement over the survival of *Sgpl1* KO mice treated with AAV-SPL. *Sgpl1* KO mice treated with AAV-SPL 2.0 exhibited improved body weight gain, correction of neurological defects, as well as minimal proteinuria up to four months of life. 

To explore the mechanisms of AAV-SPL 2.0-mediated protection against SPLIS nephrosis at the molecular level, we investigated the activation state of the STAT3 pathway, which is known to be activated by S1P signaling and has been reported to contribute to the pathogenesis of diabetic kidney disease, Alport syndrome, lupus nephritis, nephrotoxic nephritis, and polycystic kidney disease [27,28,29]. *Sgpl1* untreated KO mice display a hyper-activation of the STAT3 pathway in the kidney parenchyma and an upregulation of STAT3 target genes. However, *Sgpl1* KO mice treated in the neonatal period with AAV-SPL 2.0 showed minimal glomerular disease and almost no STAT3 pathway activation.

Despite the positive effects observed in treated KO mice during their first four months, between four and five months of age, the degree of proteinuria increased, weight steadily declined, and the animals died. Histological examination of the kidneys through PAS and IHC of treated KO mice showed more glomerulosclerosis at the time of their death. Thus, although kidney disease progression is significantly delayed, it is not completely averted by treatment with AAV-SPL 2.0. This finding suggests that targeting the kidney through improvements in capsid tropism and/or through the use of different administrative routes may be necessary to achieve substantial dose reduction and prolonged protection from SPLIS-associated nephrotic syndrome. 

Other features of SPLIS, including anemia, responded to treatment with AAV-SPL 2.0 in a similar way to the responses to AAV-SPL observed previously. Additionally, lymphopenia was attenuated in response to AAV-SPL 2.0 treatment. This is a significant advantage of AAV-SPL 2.0 over the first-generation AAV-SPL, which did not have any effect on lymphopenia. Infections represent the second leading cause of death in reported SPLIS patients. This may be explained by a combination of factors, including lymphopenia caused by the block in lymphocyte trafficking, loss of immunoglobulins in the urine in patients with nephrotic-range proteinuria, inability to breastfeed, and the vulnerability of severely ill SPLIS newborns to infection introduced by the use of peritoneal dialysis catheters, intravenous catheters, and prolonged hospital stays. Any improvement in the immunological function of SPLIS patients affords a better chance of avoiding and/or responding to infections.

The neurodevelopmental delay in pre-weaned *Sgpl1* KO mice was prevented by treatment with AAV-SPL 2.0. This suggests that the activity of AAV-SPL in the developing brains of newborn mice was not compromised by replacing the CMV promoter with a CAG promoter, the latter which has been shown to be active in the central nervous system [30]. 

Based on these overall findings, we conclude that AAV-SPL 2.0 performs at least as well as the first-generation AAV-SPL, with the advantages of a more well-established promoter element, positive effects on lymphocyte count, and longer survival times. Despite these benefits, better kidney targeting may yet be required to achieve the maximal potency and efficacy of adeno-associated viral *SGPL1* gene therapy as a potentially lifesaving targeted treatment for SPLIS.

## 4. Materials and Methods

### 4.1. Virus Construct and Packaging

pAAV-CAG-tdTomato (codon diversified) was a gift from Edward Boyden (Addgene plasmid #59462; http://n2t.net/addgene:59462 (accessed on 12 April 2019); RRID:Addgene_59462). Full-length human *SGPL1* cDNA was cloned in KpnI-EcoRI digested pAAV-CAG-tdTomato, thereby replacing the tdTomato gene in the plasmid construct. The resultant plasmid was designated AAV-SPL 2.0. To increase translational efficiency, we replaced the proline codon (position 2) of human *SGPL1* with an alanine codon that matches the Kozak consensus sequence. AAV-SPL 2.0 was sequence-verified and then amplified and packaged in AAV9 using a research-grade method suitable for in vivo use (SignaGen, Rockville, MD, USA). The rAAV was purified with 2 cycles of CsCl ultracentrifugation, resulting in >90% packaged rAAV vector and <10% empty vector. Using qPCR for rAAV titration, the virus concentration was determined to be 8 × 10^13^ vg/mL.

### 4.2. Cell Culture

Human embryonic kidney 293T (HEK293T) cells were propagated in DMEM plus 10% fetal bovine serum and passaged every third day. Cells were infected with the stated amount of virus when they were 25–30 percent confluent. The cell medium was changed the day after transfection, and cells were harvested and analyzed via Western blotting 2 days later.

### 4.3. Western Blotting and Image Quantification

Kidneys from WT, KO, and AAV-SPL 2.0-treated mice were homogenized in radioimmune precipitation assay buffer (25 mM Tris-HCl (pH 7.6), 150 mM NaCl, 1% NP-40, 1% sodium deoxycholate, and 0.1% SDS) supplemented with protease inhibitors using a Tissuelyzer and then cleared by centrifugation. These antibodies were used to detect the protein level; goat anti-human *SGPL1* (AF5535, R&D Systems, Bio-Techne, Minneapolis, MN, USA), anti-mouse SPL, rabbit anti-GAPDH (sc-25778, Santa Cruz Biotechnology, Dallas, TX, USA), anti-phosphorylated Stat3 (9145, Cell Signaling Technology, Danvers, MA, USA), anti–total STAT3 (9131, Cell Signaling Technology), Synaptopodin (SC-515842, Santa Cruz Biotechnology), and β-actin from Sigma-Aldrich (St. Louis, MO, USA). HRP-conjugated secondary antibodies [115-035-003, Goat Anti-Mouse IgG (H + L); 111-035-144, Goat Anti-Rabbit IgG (H + L), Jackson Immuno Research; sc-2020, donkey anti-goat IgG-HRP, Santa Cruz Biotechnology]. 

### 4.4. SPL Enzyme Activity

SPL activity was measured by tandem mass spectrometry using a method in which 2-hydrazinoquinoline produces a hydrazone derivative of hexadecenal product [31]. Briefly, tissues were homogenized in homogenization buffer containing 5 mM MOPS, 1 mM DTT, 1 mM EDTA, 0.25 M sucrose, and 10% glycerol with Roche protease inhibitor cocktail and Roche phosphatase inhibitor cocktail solution, followed by centrifugation at 1000× *g* for 5 min at 4 °C. The SPL enzyme reaction mixture contained 50 µL lysate of desired protein amount, 10 µL S1P (13.18 nmol), and 190 µL of reaction assay buffer (50 mM potassium phosphate buffer, pH 7.4, 1 mM EDTA, 5 mM DTT, 0.4 mM pyridoxal 5′-phosphate, 100 mM sucrose, 0.1% TritonX-100, Roche protease inhibitor cocktail, and Roche phosphatase inhibitor cocktail solution). The reaction mixture was incubated at 37 °C for 60 min. Then, 10 µL aliquot was taken and mixed with 90 µL of derivatization buffer (5 mM 2HQ, 10.28 µM (2E)-d5-hexadecenal prepared with HPLC-grade acetonitrile (5%) acidified with 70% perchloric acid). The mixture was incubated at 65 °C for 60 min. After this, 2HQ-derivatized samples were transferred into glass vials for LC-MS/MS analysis. A 5µL-aliquot was injected into the HPLC system (Agilent 1290, Santa Clara, CA, USA) equipped with a RRHD Eclipse Plus C18 column (2.1 × 50 mm, 1.8 µm Agilent) maintained at 50 °C. The column was eluted at a flow of 0.4 mL/min with a gradient of water containing 2 mM ammonium formate and 0.2% formic acid (A), and methanol containing 1 mM ammonium formate and 0.2% formic acid (B). Initial gradient condition was maintained at 65% B and a linear gradient at 65% to 100% B within 3 min and was returned to 65% at 3.10 min to allow ~2 min column re-equilibration.

### 4.5. Animals

Knockout mice in which the *Sgpl1* gene is constitutively disrupted (*Sgpl1* KO mice) have been described previously [21]. Heterozygous (HET) knockout breeders were mated, and newborns were genotyped by toe biopsy on day of life (DOL) 1. On DOL 1–2, *Sgpl1* knockout pups were anesthetized with isoflurane and injected intravenously with 20 µL of PBS or AAV-SPL 2.0 in PBS, as we described previously [14]. Unless otherwise specified, the dose of 5 × 10^11^ vg was used. In the final experiment, doses ranging from 3 × 10^10^ to 3 × 10^12^ in half-log increments were used. Virus solution was injected into the superficial temporal vein using a Yale Model YA-12 syringe pump. Mice were monitored for health and weighed weekly. Treated and untreated *Sgpl1* knockout mice, and wild type (WT) and HET littermates, were evaluated for neurodevelopment starting on DOL 11 by scoring for six developmental milestones, as we described [14]. Twenty-four urine collections were performed in metabolic chambers and analyzed for urine creatinine and albumin (UC Davis Comparative Pathology Laboratory). When a mouse was moribund, it was euthanized by CO_2_ inhalation, and that day was considered the day of death. Terminal phlebotomy and tissue harvest were performed.

### 4.6. Tissue Collection and Histology

All tissues were divided into four equal parts in an anatomically consistent manner (i.e., 2 kidneys bisected). One sample was placed into RNA later and incubated at 4 °C overnight, then frozen at −80 °C for qRT-PCR, one flash frozen in LN_2_ for SPL assays, and one placed in formalin for histology investigation. Periodic acid–Schiff staining was performed on 3 µm paraffin sections and analyzed and scored blind to treatment. The total number of glomeruli was recorded, along with number of globally sclerosed glomeruli. Qualitative assessment of segmentally sclerosed glomeruli, glomeruli with mesangial hypercellularity, and the proportion of interstitium with interstitial fibrosis was carried out by a nephropathologist. For immunohistochemistry, mouse kidney sections (3 µm) were dewaxed, rehydrated, and subjected to heat-induced antigen retrieval by incubating in Tris-EDTA buffer (pH 9.0, Abcam ab93684, Cambridge, UK) in 95 °C water bath for 1 h. Endogenous peroxidase was blocked with 3% H_2_O_2_ (Sigma H1009, St. Louis, MO, USA) in Tris-buffered saline (TBS, pH 7.4) for 30 min. And then, sections were blocked with 10% normal goat serum in 5% BSA at room temperature for 2 h. A phospho-STAT3 (Tyr705) (D3A7) XP^®^ rabbit mAb (Cell Signaling, 9145, Danvers, MA, USA) was used at a concentration of 1:200 at cold room temperature (4 °C) overnight. Antibody binding was detected using secondary antibody, goat anti-rabbit Ig-HRP (1:100, Southern Biotech, 4010-05), and betazoid DAB chromogen kit (Biocare Medical, BDB2004, Pacheco, CA, USA). Counterstaining was performed using hematoxylin (Fisher Healthcare, 220-102, Waltham, MA, USA). All images were captured using a Nikon Eclipse 80i microscope equipped with a digital camera (Nikon DS-Ri1, Tokyo, Japan).

### 4.7. Hematological Testing

Whole blood for a complete blood count (CBC) was placed into dipotassium ethylenediamine tetra acetic acid (EDTA) and analyzed within 1–3 h after blood collection. Collected whole blood specimens were stored at 4 °C or room temperature before experiment. Blood was diluted in 80 μL of CellPak (Sysmex America, Lincolnshire, IL, USA). A total of 100 μL diluted blood samples were analyzed using the XT-2000iV veterinary hematology analyzer (Sysmex, Kobe, Japan) to record a full standard hematology profile.

### 4.8. Urine ACR Analyses

Serum and urine albumin and creatinine were measured using the COBAS INTEGRA 400 plus instrument (Roche Diagnostics, Indianapolis, IN, USA) by the University of California Davis Comparative Pathology Laboratory (Davis, CA, USA).

### 4.9. Quantitative RT-PCR

Total RNA was extracted by TRIzol reagent (Thermo Fisher Scientific, Waltham, MA, USA). DNase was used to remove genomic DNA contamination. A total of 2 μg of DNase I– treated RNA was used for the first-strand cDNA synthesis using the SuperScript III reverse transcriptase (Life Technologies (Carlsbad, CA, USA), Thermo Fisher Scientific). Quantitative real-time PCR was performed using PowerUp SYBR Green Master Mix in QuantStudio™ 6 7 Flex Real-Time PCR System (Life Technologies, Thermo Fisher Scientific). Human and murine *SGPL1* mRNA and levels of cytokines were measured. Cq values of samples were normalized to the corresponding Cq values of Actb. Quantification of the fold change in gene expression was determined by the comparative Cq method. Primers are listed in Table 1.

### 4.10. Sphingolipid Quantitation

Blood was placed into EDTA tubes, spun at 4 °C to separate plasma, which was collected and frozen. Frozen tissues were bead homogenized in Tris-buffer. Homogenates were spiked with internal standard. Using targeted metabolomics, we measured 54 sphingolipid species in plasma and tissue using single-reaction monitoring with an Agilent 6495C QQQ-LC-MS/MS coupled with a 1290 Infinity UPLC system, as we described [32]. 

### 4.11. Statistical Analyses

For comparing 2 groups of equal size, unpaired students *t*-test was used. For groups of ≥3, one-way ANOVA was used. *p* < 0.05 was considered significant. When multiple comparisons were made, Bonferroni’s correction was performed. Two-tailed *t* tests were performed in all cases.

## Figures and Tables

**Figure 1 ijms-24-15560-f001:**
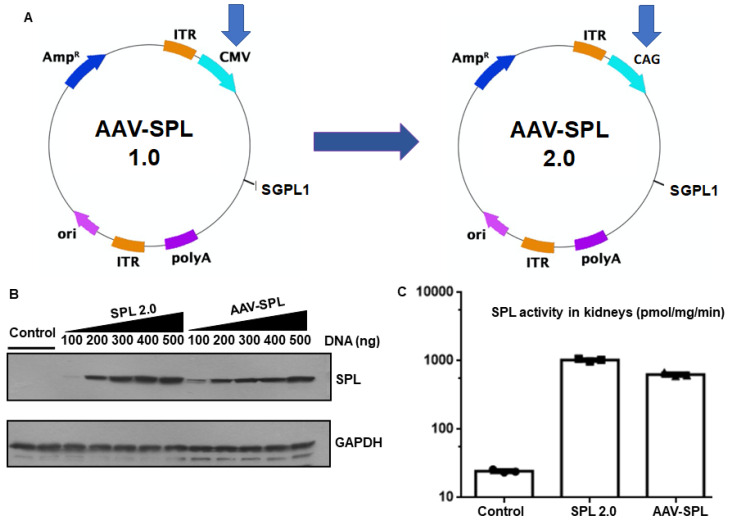
AAV-SPL 2.0 in vitro activity is equal or greater than AAV-SPL. (**A**) Schematic design of AAV-SPL 1.0 (hereafter referred to as AAV-SPL) and AAV-SPL 2.0. Human *SGPL1* cDNA (hSPL) was cloned into AAV vectors using CMV (1.0) or CAG (2.0) promoters and used to transduce HEK293T cells; (**B**) Western blotting was performed using an hSPL antibody from the whole-cell extract of cells treated with vehicle or with 100–500 nanogram (ng) AAV-SPL 2.0 or AAV-SPL DNA; and (**C**) SPL activity in control, AAV-SPL 2.0, and AAV-SPL-treated extracts (cells transfected with 500 ng DNA were used for the activity assay). *p* = 0.003 between AAV-SPL 2.0 and AAV-SPL.

**Figure 2 ijms-24-15560-f002:**
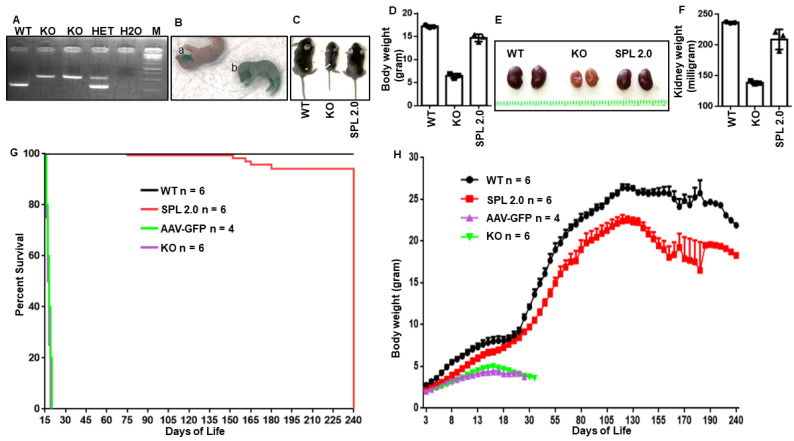
Treatment of 3-day-old (DOL3) *Sgpl1* KO pups with AAV-SPL 2.0 improved kidney and body weight and prolonged survival. (**A**) PCR using DNA from wild-type (WT), *Sgpl1* knockouts (KO), and heterozygous (HET) pups verify the successful disruption of the *Sgpl1* gene in KO pups; (**B**) injection of AAV-SPL 2.0 to DOL 3 pups (a: unsuccessful injection and b: successful injection) after confirmation of genotyping; (**C**) and size discrepancy between WT, untreated *Sgpl1*-KO, and AAV-SPL 2.0-treated *Sgpl1*-KO littermates at 26 DOL was observed. (**D**) Body weight quantified at 26 DOL; for WT vs. KO, *p* < 0.0001; for AAV-SPL 2.0 vs. untreated KO, *p* < 0.0076; and for WT vs. treated KO, no significant difference (NSD). (**E**) Representative images of kidneys from 26 DOL WT, untreated *Sgpl1*-KO, and AAV-SPL 2.0 mice. (**F**) kidney weights of 26 DOL WT, KO, and AAV-SPL 2.0-treated mice; for AAV-SPL 2.0 vs. KO, *p* < 0.0019. (**G**) Kaplan–Meier survival curve for WT, KO, and AAV-SPL 2.0-treated mice. Log-rank test with Bonferroni’s correction: for WT vs. all other groups, *p* < 0.0001; AAV-SPL 2.0 vs. KO, *p* < 0.0001; and KO vs. AAV-GFP, NSD. (**H**) Weight gain of WT, untreated KO, AAV-GFP-treated KO, and AAV-SPL-2.0-treated KO mice.

**Figure 3 ijms-24-15560-f003:**
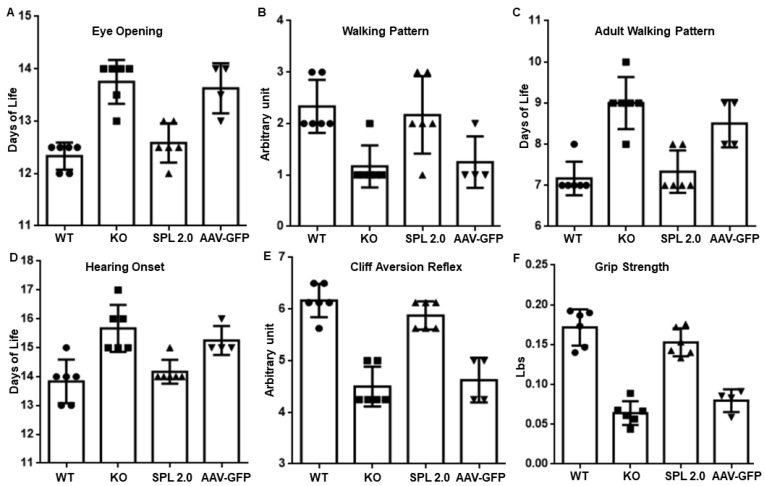
AAV-SPL 2.0 treatment prevents neurodevelopmental delay in pre-weaned *Sgpl1* KO mice. (**A**–**F**) Six neurodevelopmental milestones were recorded in WT (*n* = 6), KO (*n* = 6), AAV-SPL 2.0-treated KO (*n* = 6) and AAV-GFP-treated KO (*n* = 4) pups. For (**A**,**E**), an unpaired t-test with Bonferroni’s correction was used. For (**B**–**D**,**F**), a Mann–Whitney U test with Bonferroni’s correction was used. For WT vs. KO comparisons: (**A**, *p* < 0.0001); (**B**, *p* = 0.0129); (**C**, *p* = 0.0001); (**D**, 0.001); (**E**, *p* = 0.0001); and (**F**, *p* = 0.0001). There was NSD between AAV-SPL 2.0 vs. WT and KO vs. GFP for (**A**–**F**).

**Figure 4 ijms-24-15560-f004:**
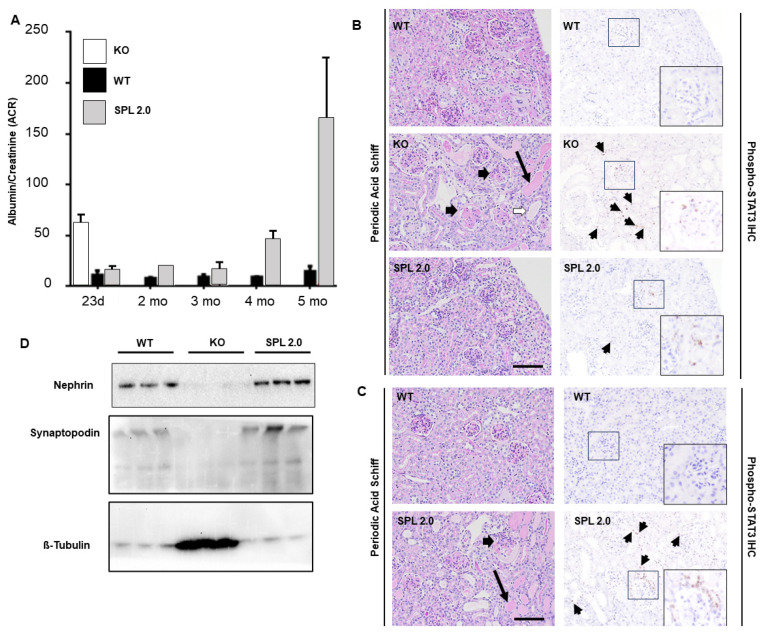
AAV-SPL 2.0 treatment delays onset of nephrotic syndrome in *Sgpl1* KO mice. (**A**) Urine ACR in WT (black bars, *n* = 3), KO (white bar, *n* = 3), and AAV-SPL-treated KO (gray bar, AAV, *n* = 3) mice at mentioned time periods. At day 23, WT vs. KO, *p* < 0.0001 and WT vs. AAV-SPL 2.0 was NSD. At the period of 2 months and 3 months, WT vs. AAV-SPL 2.0 was NSD, whereas at the period of 4 months and 5 months, WT vs. AAV-SPL 2.0, *p* < 0.0018 and 0.014, respectively. (**B**, left panel) 26-days-old WT, KO, and AAV-SPL 2.0 kidney sections stained with periodic acid–Schiff. KO kidney sections show diffuse mesangial expansion by cells and matrix (black arrowhead), as well as moderate scattered protein casts (black arrow) and acute tubular injury (white arrowhead), none of which are seen in sections of WT or AAV-SPL 2.0-treated KO kidneys. These images are representative of 3 mice per group. (**B**, right panel) p-STAT3 IHC of WT, KO, and AAV-SPL 2.0-treated KO kidney sections. Multifocal positive nuclei are seen in cortex, tubules, glomeruli, and probably some immune cells (insert and black arrowheads) in KO kidney sections, whereas only rare signals were observed in AAV-SPL 2.0-treated KO mice sections (insert and black arrowhead), scale bar = 100 µm; (**C**, left panel) at 5 months of age, no untreated KO mice were alive. Five-month-old WT and AAV-SPL 2.0-treated KO mouse kidney sections stained with periodic acid–Schiff (left) and p-STAT3 IHC (right). AAV-SPL 2.0-treated KO kidney sections displayed severe interstitial fibrosis (black arrowhead), scattered protein casts (black arrow), and acute tubular injury. (**C**, right panel) Treated KO mice kidneys show more p-STAT3 signals (insert and black arrowheads), whereas WT mice kidneys showed no p-STAT3 signal or pathology; scale bar = 100 µm. (**D**) Immunoblot showing Nephrin and Synaptopodin in kidneys of WT, KO, and AAV-SPL 2.0-treated KO mice; *n* = 3/group. β-Tubulin is a loading control. Note: KO samples were massively overloaded to demonstrate complete absence of podocyte marker protein expression (see corresponding Tubulin band).

**Figure 5 ijms-24-15560-f005:**
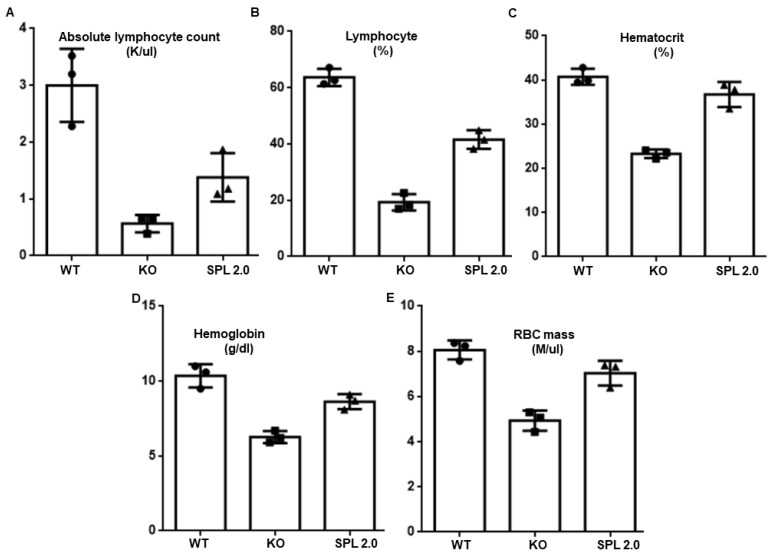
Hematological parameters of WT, AAV-SPL 2.0-treated and untreated *Sgpl1* KO mice. (**A**–**E**) Blood parameters including absolute lymphocyte count (K/μL), percentage lymphocytes (% lymph), hematocrit hemoglobin (g/dL), and RBC mass (M/μL), (%) were investigated in WT, KO, and AAV-SPL 2.0-treated KO. All mice were euthanized at 26 DOL. All parameters were evaluated using an unpaired two-tailed t-test with Bonferroni’s correction. For absolute lymphocyte count, WT vs. KO, *p* = 0.003; AAV-SPL 2.0 vs. WT, *p* = 0.022; and AAV-SPL 2.0 vs. KO, *p* = 0.03. For percentage lymphocytes, WT vs. KO, *p* < 0.0001; AAV-SPL 2.0 vs. WT, *p* < 0.001; and AAV-SPL 2.0 vs. KO, *p* = 0.009. For hematocrit, WT vs. KO, *p* = 0.0001, and AAV-SPL 2.0 vs. KO, *p* = 0.0015. For hemoglobin, WT vs. KO, *p* = 0.001; AAV-SPL 2.0 vs. WT, *p* = 0.031; and AAV-SPL 2.0 vs. KO, *p* = 0.0032. For RBC mass, WT vs. KO, *p* = 0.009; AAV-SPL 2.0 vs. WT, NSD; and AAV-SPL 2.0 vs. KO, *p* = 0.0069.

**Figure 6 ijms-24-15560-f006:**
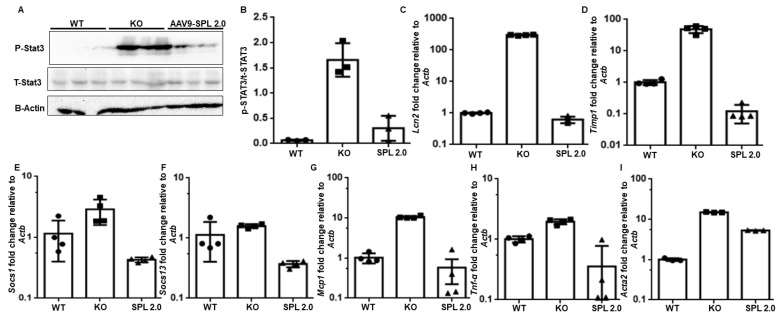
Attenuation of STAT3 activation and cytokines in kidneys of AAV-SPL 2.0-treated *Sgpl1* KO mice. (**A**) Immunoblot showing total and phosphorylated (tyrosine 705) STAT3 in kidneys of WT, KO, and AAV-SPL 2.0-treated KO mice; *n* = 3/group. B-ACTIN is a loading control. (**B**) Quantification of immunoblot (**A**). WT vs. KO, *p* < 0.001, and WT vs. AAV-SPL 2.0, NSD; (**C**–**I**) Relative expression of STAT3 target genes *Lcn2*, *Timp1*, *Socs3*, *Socs1* and cytokines *Mcp1*, *Tnf-α*, and *Acta2* in kidneys of WT, KO, and AAV-SPL 2.0-treated KO mice (*n* = 4). For *Lcn2* and *Timp1*, WT vs. KO, *p* < 0.0001. For *Socs3* and *Socs1*, AAV-SPL 2.0 vs. KO, *p* < 0.0001 and 0.0090, respectively. For *Tnf-α* and *Acta2*, AAV-SPL 2.0 vs. KO, *p* < 0.0001. There was NSD between WT and AAV-SPL 2.0 for all genes except *Acta2* (*p* < 0.001).

**Figure 7 ijms-24-15560-f007:**
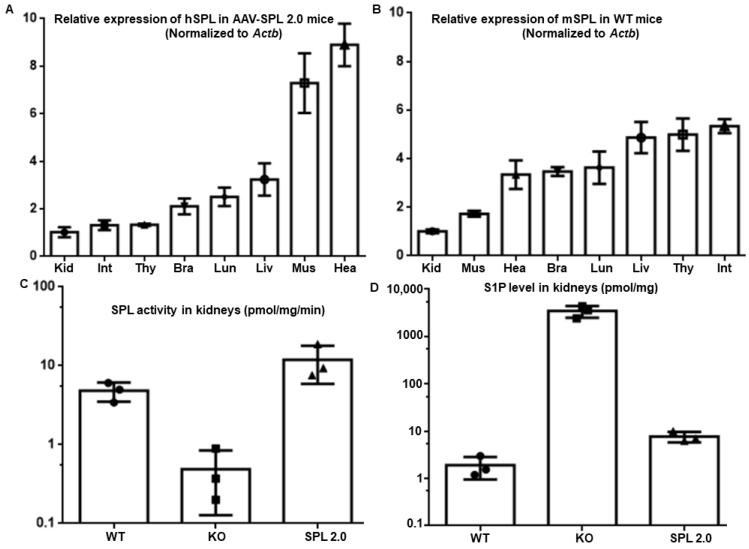
Bioavailability of AAV-SPL 2.0 and impact on kidney SPL enzyme activity. (**A**) Relative hSPL (*SGPL1*) levels in different tissues of AAV-SPL 2.0-treated KO mice (*n* = 4); (**B**) relative expression of mSPL (*Sgpl1*) in tissues of WT mice (*n* = 4); and (**C**) SPL activity level in kidney WT, KO, and AAV-SPL 2.0-treated KO mice. Student’s *t*-test for WT vs. KO, *p* < 0.03, and AAV vs. KO, *p* < 0.02; *n* = 3 per group. (Kid: Kidney, Int: Intestine, Thy: Thymus, Bra: Brain, Lun: Lung, Liv: Liver, Mus: Muscle, and Hea: Heart). (**D**) S1P levels in the kidney tissues of WT, KO, and SPL 2.0-treated KO mice. WT vs. KO, *p* = 0.0037; AAV-SPL 2.0 vs. KO, *p* = 0.0037; and AAV-SPL 2.0 vs. WT, *p* = 0.009.

**Figure 8 ijms-24-15560-f008:**
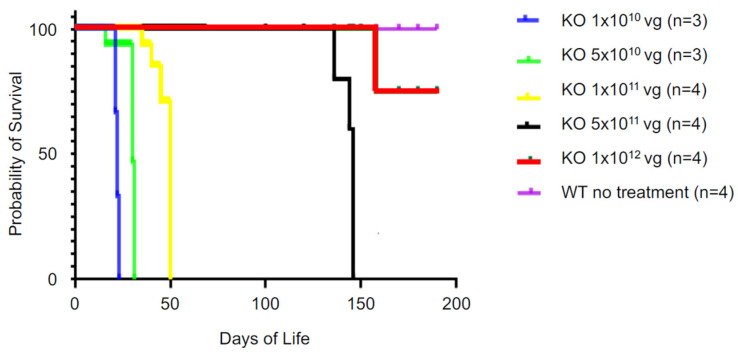
In vivo potency of AAV-SPL 2.0. Kaplan–Meier curve showing the survival probability of mice given AAV-SPL 2.0 at five different doses.

**Table 1 ijms-24-15560-t001:** Primers used for qRT-PCR.

Primer Name	Sequence (5′-3′)
h-*SGPL1*-F	CAAGACCAAGGATGATATTAGC
h-*SGPL1*-R	CAGAAGGCGTCCATAGAG
m-*Sgpl1*-F	TTTCCTCATGGTGTGATGGA
m-*Sgpl1*-R	CCCCAGACAAGCATCCAC
m*MCP1*-F	TTAAAAACCTGGATCGGAACCAA
m*MCP1*-R	GCATTAGCTTCAGATTTACGGGT
m*Lcn2*-F	TGGCCCTGAGTGTCATGTG
m*Lcn2*-R	CTCTTGTAGCTCATAGATGGTGC
m*Socs1*-F	CTGCGGCTTCTATTGGGGAC
m*Socs1*-R	AAAAGGCAGTCGAAGGTCTCG
m*Socs3*-F	CCCTTGCAGTTCTAAGTTCAACA
m*Socs3*-R	ACCTTTGACAAGCGGACTCTC
m*Timp1*-F	GCAACTCGGACCTGGTCATAA
m*Timp1*-R	CGGCCCGTGATGAGAAACT
m*Tnf-α*-F	CAGGCGGTGCCTATGTCTC
m*Tnf-α*-R	CGATCACCCCGAAGTTCAGTAG
m*Acta2*-F	GCATCCACGAAACCACCTAT
m*Acta2*-R	ATCTCCTTCTGCATCCTGTC
m-B-Actin-F	GGCTGTATTCCCCTCCATCG
m-B-Actin-R	CCAGTTGGTAACAATGCCATGT

## Data Availability

All data are included in the article.

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
