# Peer review of "AAV-SPL 2.0, a Modified Adeno-Associated Virus Gene Therapy Agent for the Treatment of Sphingosine Phosphate Lyase Insufficiency Syndrome"

_ijms, 2023, doi:10.3390/ijms242115560_

Round 1
Reviewer 1 Report
SPLIS is a rare devastating inborn-error metabolic disease caused by mutations in the SGPL gene. So far no therapeutic options are available for patients. The authors present an improved and promising gene therapy for SPLIS patients. The study was excellently designed and performed. The data is clearly described and carefully discussed. The results are of a high scientific and clinical importance. The manuscript can be accepted in the present form.
Author Response
Thank you very much for appreciating our study.
Reviewer 2 Report
In a very interesting study, the authors describe gene therapy of SPLIS in KO mice using gene modifications of the original virus. With this type of therapy, they achieved longer survival of mice affected with such a metabolic error. However, as in all animal models, it is not ultimately clear if results are applicable to humans, although in my personal opinion this probability is rather high. For better understanding of abbreviations used, generation of a list of abbreviations would be helpful. The paper is fit for publication in the special section after minor corrections.
Additional comments:
In general English is fine, but single typing errors should be eliminated. (e.g. a missing space between RNA and later (RNAlater) in the MM section. Careful proof readimg is required.
Author Response
Comment 1: For better understanding of abbreviations used, generation of a list of abbreviations would be helpful.
Response: According to the journal style, we are requested to abbreviate after first instance of usage and not include a separate abbreviations section. However, we have carefully edited the manuscript to ensure that all abbreviations have been properly defined at the first instance.
Comment 2: The paper is fit for publication in the special section after minor corrections...In general English is fine, but single typing errors should be eliminated. (e.g. a missing space between RNA and later (RNAlater) in the MM section. Careful proof reading is required.
Response: Thank you for pointing out our typos mistake. We have corrected all typographical errors and removed excess spaces in the revised version.
Reviewer 3 Report
Review report for Manuscript ijms-2645874
A summary
The design of the study was relatively well conducted. It could be said that there is enough evidence to conclude that AAV-SPL 2.0 is efficient in treating sphingosine-1-phosphate lyase insufficiency syndrome (SPLIS) in mice.
Considering the importance of what the authors confirm with their results, I think that the discussion is inadequate and that the results are not sufficiently discussed and explained.
Figure 1. and Figure 4. The results of immunoblotting and immunohistochemistry are not representative. Western blots are quite bad for publication.
Lines 76- 78 The authors stated: However, subsequent dose range finding studies revealed that a minimum dose equivalent of 3-5e14 vector genomes/kg was required to achieve long-term survival of Sgpl1 knockout mice (our unpublished data), a dose that has been associated with severe toxicity and deaths in some human clinical trials.
What is the dose the authors used in the study? Is the applied dose safe for future human studies? How would extrapolation of the dose from mice to humans be done?
Author Response
Comment 1: Considering the importance of what the authors confirm with their results, I think that the discussion is inadequate and that the results are not sufficiently discussed and explained.
Response: Thank you for suggesting we improve the discussion part of the manuscript. We have extensively modified the discussion, emphasizing the most important observations from our study and bolstering our conclusions with the relevant references regarding discoveries in the literature.
Comment 2: Figure 1. and Figure 4. The results of immunoblotting and immunohistochemistry are not representative. Western blots are quite bad for publication.
Response: Thank you for suggesting improving immunoblotting and IHC. We have provided new images for Synaptopodin WB. Specifically, in the revised image of western blotting results depicted in Figure 4D, we have loaded more KO lysate to make it absolutely clear that no Synaptopodin can be detected even at higher concentration in KO mice kidneys (see corresponding B-Tubulin level in KO lanes). Moreover, we provided better quality images for IHC, using lighter images to emphasize the STAT3 IHC signal in the cells. We further provided arrowheads to show the significant amount of dispersed signal in KO kidneys compared to treated KO kidneys at early time points, and the positive signal present in treated KO kidneys at late time points. We believe that the combination of the inset with the arrowheads demonstrate the differences clearly now. Regarding Figure 1 western blot, we think this is very clear and representative as well as being very consistent with the SPL activity assay results, so we have not altered this image.
Comment 3: Lines 76- 78 The authors stated: However, subsequent dose range finding studies revealed that a minimum dose equivalent of 3-5e14 vector genomes/kg was required to achieve long-term survival of Sgpl1 knockout mice (our unpublished data), a dose that has been associated with severe toxicity and deaths in some human clinical trials. What is the dose the authors used in the study? Is the applied dose safe for future human studies? How would extrapolation of the dose from mice to humans be done?
Response: The dose we used is 5e11 vg = 3.3e14 vg/kg. The dose is equivalent to the higher end doses used in human AAV gene therapy clinical trials. However, to reduce the chance of toxicity of our agent, we are pursuing methods (as described in Discussion section) to improve kidney targeting prior to advancing the therapy to the clinic.
Round 2
Reviewer 3 Report
The authors have significantly improved the manuscript and responded to I comments. I suggest that you accept the manuscript in its current form for publication.